# Glycaemic Control in Patients Undergoing Percutaneous Coronary Intervention: What Is the Role for the Novel Antidiabetic Agents? *A Comprehensive Review of Basic Science and Clinical Data*

**DOI:** 10.3390/ijms23137261

**Published:** 2022-06-30

**Authors:** Annunziata Nusca, Francesco Piccirillo, Federico Bernardini, Aurelio De Filippis, Federica Coletti, Fabio Mangiacapra, Elisabetta Ricottini, Rosetta Melfi, Paolo Gallo, Valeria Cammalleri, Nicola Napoli, Gian Paolo Ussia, Francesco Grigioni

**Affiliations:** 1Unit of Cardiac Sciences, Department of Medicine, Campus Bio-Medico University of Rome, 00128 Rome, Italy; f.piccirillo@unicampus.it (F.P.); f.bernardini@unicampus.it (F.B.); a.defilippis@unicampus.it (A.D.F.); federica.coletti@unicampus.it (F.C.); f.mangiacapra@policlinicocampus.it (F.M.); e.ricottini@policlinicocampus.it (E.R.); r.melfi@policlinicocampus.it (R.M.); p.gallo@policlinicocampus.it (P.G.); v.cammalleri@policlinicocampus.it (V.C.); g.ussia@policlinicocampus.it (G.P.U.); f.grigioni@policlinicocampus.it (F.G.); 2Unit of Endocrinology and Diabetes, Department of Medicine, Campus Bio-Medico University of Rome, 00128 Rome, Italy; n.napoli@policlinicocampus.it

**Keywords:** glycaemic control, hyperglycaemia, glycaemic variability, diabetes mellitus, coronary artery disease, percutaneous coronary intervention, coronary stenting, anti-diabetic agents

## Abstract

Coronary artery disease (CAD) remains one of the most important causes of morbidity and mortality worldwide, and revascularization through percutaneous coronary interventions (PCI) significantly improves survival. In this setting, poor glycaemic control, regardless of diabetes, has been associated with increased incidence of peri-procedural and long-term complications and worse prognosis. Novel antidiabetic agents have represented a paradigm shift in managing patients with diabetes and cardiovascular diseases. However, limited data are reported so far in patients undergoing coronary stenting. This review intends to provide an overview of the biological mechanisms underlying hyperglycaemia-induced vascular damage and the contrasting actions of new antidiabetic drugs. We summarize existing evidence on the effects of these drugs in the setting of PCI, addressing pre-clinical and clinical studies and drug-drug interactions with antiplatelet agents, thus highlighting new opportunities for optimal long-term management of these patients.

## 1. Introduction

Coronary artery disease (CAD) still represents a leading cause of cardiovascular death, alongside cerebrovascular diseases. Since Andreas Grüntzig performed the first percutaneous coronary angioplasty in 1977, percutaneous coronary intervention (PCI) has taken off, becoming a milestone in the treatment of CAD. Moreover, in the last 40 years, revolutionary changes in technologies such as new-generation drug-eluting stents (DES), intracoronary imaging modalities, and debulking strategies have resulted in a significant widening in PCI indications and improved clinical success, such that most revascularization procedures are currently performed percutaneously [1,2]. 

Despite these significant improvements, patients with diabetes mellitus (DM) still have worse clinical outcomes following PCI than non-diabetics [3,4]. DM represents an important risk factor for most of the intra-procedural and post-procedural stent-related complications such as periprocedural myocardial infarction (PMI), stent thrombosis (ST), in-stent restenosis (ISR), neo-atherosclerosis, as well as accelerates atherosclerosis progression in other coronary segments [5]. Moreover, hyperglycaemia and increased glycaemic variability (GV) have been reported to be independent predictors of poor outcomes after PCI, regardless of a prior history of known DM and specifically in the setting of acute coronary syndromes (ACS) [6,7]. 

Nowadays, a new paradigm shift in diabetes treatment occurs since clear evidence of cardiovascular benefits associated with novel antidiabetic drugs has been reported in several randomized trials [8,9,10]. Their cardiovascular effectiveness has been primarily documented in preventing heart failure (HF) and cardiovascular mortality. Given their anti-proliferative, anti-thrombotic and anti-inflammatory effects, these agents might also greatly benefit patients undergoing PCI.

This review aims to provide an overview of the pathophysiological mechanisms linking hyperglycaemia with short- and long-term PCI complications. We also sought to investigate the potential contrasting actions of new antidiabetic drugs, summarizing existing pre-clinical and clinical evidence on the effects of these drugs in the setting of PCI.

## 2. Mechanisms of Hyperglycaemia-Induced Vascular Damage

In patients undergoing PCI, hyperglycaemia and GV, irrespective of the presence of known DM, have been demonstrated to induce an increase in reactive oxygen species (ROS) production, inflammation, endothelial dysfunction, and platelet reactivity, thus favouring the progression of atherosclerotic plaque and increasing the risk of in-stent complications such as thrombosis and restenosis [11] (Figure 1).

### 2.1. Increased Oxidative Stress and PKC-Mediated Pathway

Chronic hyperglycaemia and even more GV are associated with increased production of ROS, including free radicals such as superoxide anion (O^2−^), lipid radicals (ROO^−^), hydroxyl radical (HO^−^), and not free radicals such as hydrogen peroxide (H_2_O_2_), hypochlorous acid (HClO) and peroxynitrite (ONOO^−^) [12]. In humans, ROS are mainly produced in the mitochondrial respiratory chain by several enzymes, including xanthine-oxidase (XO), NADPH-oxidase (NOX), and uncoupled nitric oxide synthase (NOS) [13]. Glucose fluctuations have been reported to influence these enzymes inducing an increased ROS production through the activation of different pathways: protein kinase C (PKC), protein kinase B (PKB or AKT), mitogen-activated protein kinase (MAPK) and nuclear factor kappa-light-chain-enhancer of activated B cells (NF-kB) [14]. Among these, the activation of PKC has emerged as a crucial mechanism. Interestingly, different molecular pathways can induce PKC activation. In hyperglycaemic conditions, glucose is converted to polyalcohol sorbitol through the polyol pathway, resulting in an increased intracellular NADH/NAD+ ratio and an enhanced formation of diacylglycerol (DAG). As a result, increased DAG levels provoke PKC activation. Furthermore, PKC’s domain also binds Ca^2+^; thus, the G-protein coupled receptor (GPCR)-mediated cleavage of the phosphatidylinositol 4,5-biphosphate (PIP2) in inositol-1,4,5-triphosphate (IP3) and Ca^2+^, activates IP3 receptors on the smooth endoplasmic reticulum (ER) and facilitates the release of intracellular calcium stores with subsequent activation of PKC [15]. PKC is also directly activated by intracellular ROS, such as by O^2−^, overproduced in hyperglycaemic conditions [16]. Interestingly, hyperglycaemia-induced PKC activation has been reported to have significant intracellular and intercellular consequences, affecting endothelial permeability, vasoconstriction, extracellular matrix synthesis/turnover, cell growth, cytokine activation and leukocyte adhesion, thus emerging as a potential therapeutic target to prevent diabetic vascular complications [17]. Furthermore, high glucose levels induce the activation of the hexosamine pathway. The resulting formation of uridine diphosphate N-acetylglucosamine (UDP-GlcNAc) leads to the glycosylation of several intracellular proteins that regulate pro-inflammatory and pro-thrombotic genes [18]. Hyperglycaemia also stimulates the formation of advanced glycation end products (AGEs), which linking to their receptors (RAGE), induce the generation of intracellular ROS and the subsequent activation of the redox-sensitive transcription factor NF-kB, which modulates the expression of a variety of genes associated with inflammation and atherosclerosis [18]. On the other hand, hyperglycaemia is also characterized by an abnormal imbalance of antioxidant defences. In this regard, the antioxidant enzyme superoxide dismutase (SOD) concentration, which usually helps to maintain ROS levels under a certain threshold, is reduced in hyperglycaemic conditions [17,18]. Among all responsible mechanisms for this latter pathophysiological mechanism, miRNA-21 induced dysregulation of Krev/Rap1 interaction trapped-1 (KRIT1), extracellular signal-regulated kinase (ERK) and nuclear erythroid 2 related factor-2 (NFE2L2 or NRF2) signalling has been recently investigated [19]. 

### 2.2. Endothelial Function

Endothelial cells play a central role in the pathogenesis of atherosclerosis and are strongly influenced by hyperglycaemia and glucose fluctuations [20]. Quagliaro et al. demonstrated that exposure to intermittent high glucose levels in human umbilical vein endothelial cells (HUVEC) produces endothelial cell dysfunction, apoptosis, and ROS production [21]. Glucose variations have also been demonstrated to influence the production of nitric oxide (NO) by endothelial cells and subsequent angiogenesis [22]. Therefore, Biscetti et al. demonstrated that GV in diabetic mice inhibits vascular endothelial growth factor (VEGF), endothelial NOS and AKT after ischemic injury [23]. The consequence of e-NOS downregulation is a reduction of NO synthesis and impaired endothelial-dependent vasodilation. Additionally, lower VEGF levels reduce angiogenesis by inhibiting angiogenic sprouting after ischemia [24]. 

Moreover, in hyperglycaemic conditions, the short NO remainder reacts with O^2−^ to form ONOO-, a strong cytotoxic oxidant. This latter compound has a double effect: direct damage to protein, lipids, DNA, and e-NOS enzyme uncoupling through oxidation of tetrahydrobiopterin [25,26]. Whether coupled e-NOS uses L-arginine to produce L-citrulline and NO, favouring cyclic guanosine monophosphate (cGMP) synthesis in vascular smooth muscle cells (VSMCs) and vasodilation, uncoupled e-NOS enhances O^2−^ rather than NO with subsequent cellular damage [27]. 

In turn, O^2−^ also stimulates the formation of AGEs [28]. AGEs, molecules generated by the non-enzymatic reaction between proteins, lipids or nucleic acids with the aldehydic group of sugars, cause vascular damage and accelerated atherosclerotic plaque formation through RAGE binding [29,30].

In endothelial cells, PKC βI and βII isoform activation induced by intermittent high glucose can also increase adhesion molecule expression. HUVECs exposed to stable and intermittent high glucose conditions show an increased expression of intercellular adhesion molecule-1 (ICAM-1), vascular cell adhesion molecule-1 (VCAM-1) and E-selectin [31]. This event may promote monocyte adhesion to endothelium, their migration in sub-endothelial space and consequent differentiation in foam cells [32]. 

Finally, hyperglycaemia and GV can induce apoptosis and autophagy in endothelial cells. HUVECs exposed to intermittent high glucose show elevated levels of p53 with consequent p21, p53 upregulated mediator of apoptosis (PUMA), phosphatase and tensin homologue (PTEN), TP53-induced glycolysis and apoptosis regulator (TIGAR) overexpression. These proteins are involved in cell growth arrest and apoptosis [33]. These fluctuations increase miR-1273g-3p levels and subsequent autophagy in the same cell line [34].

### 2.3. Vascular Smooth Muscle Cells

Glucose fluctuations significantly influence the proliferation and migration of VSMCs, which play a central role in atherosclerosis progression and ISR. Indeed, GV promotes lipid-rich plaques with thin fibrous caps and neointimal thickening independently of dyslipidaemia control [35,36]. These data were confirmed by a recent metanalysis that underlines how reducing glucose fluctuations correlates with improved intimal-media thickness [37]. 

Hyperglycaemia influences VSMCs migration and proliferation through different molecular pathways. Sung Hoon Yu et al. showed that intermittent hyperglycaemia results in the accumulation of VSMCs mediated by MAPK, big mitogen-activated protein kinase 1 (BMK1), phosphoinositide 3-kinases (PI3K), and NF-κB [38]. This process is also driven by miRNA21, miRNA146a, matrix metalloprotease-2 (MMP-2) and osteopontin (OPN) [39,40]. Moreover, high glucose levels have been demonstrated to downregulate the insulin receptor substrate-1 (IRS-1), thus decreasing the p53/Krüppel-like factor 4 (KLF-4) association and enhancing VSMCs dedifferentiation and proliferation [41]. Concordantly, the VSMCs expression of genes participating in ERK mitogenic response is activated in the setting of hyperglycaemia, which furtherly stimulates cell proliferation [41]. Hyperglycaemia-induced ROS production also favours lipid peroxidation, affecting ion transport across cell membranes; among channels and pump altered, an abnormal function of the sarcoplasmic/endoplasmic reticulum ATPase (SERCA) results in modified intracellular Ca^2+^ signalling in VSMCs, enhancing their migration [42]. On the other side, hyperglycaemia inhibits apoptosis of VSMCs by upregulation of several anti-apoptotic proteins such as Bcl2, Bcl-xl and Bfl-1/A1 [43].

### 2.4. Inflammation

Atherosclerotic plaque formation and progression are favoured by monocyte adhesion to vascular endothelium and their subsequent migration in the sub-intimal layer. CD14+ and CD16+ sub-groups are the most represented in patients with CAD and peripheral artery disease. A recent virtual histology study showed that CD14+ and CD16+ monocyte accumulation correlates with plaque instability; most importantly, GV seems to increase their expression leading to accelerated atherosclerosis [44]. Consistently, monocytes exposed to an acute glucose load show a significant and rapid increase (within 120 min) of the adhesion molecule Mac-1 (CD11b), which interacts with endothelial ICAM-1 and with non-endothelial matrix ligands favouring sub-endothelial cellular migration [45]. Mac-1 also functions as a link between cellular adhesion and thrombosis via the activation of factor X and the coagulant cascade [46]. Other hyperglycaemia-stimulated receptors involved in monocyte adhesion are leukocyte function antigen-1 (LFA-1 or CD11a) and ICAM-1 [45]. 

Furthermore, an acute increase in blood glucose concentrations causes a rapid rising in several cytokine levels, such as interleukin 6 (IL-6), interleukin 18 (IL-18) and tumour necrosis factor-alpha (TNF-alpha), contributing to creating a pro-inflammatory environment [47]. Hyperglycaemia also induces the inactivation of CD59, an extracellular cell-membrane regulatory protein that inhibits the assembly of the membrane attack complex (MAC). The CD59 inactivation and the following increase in MAC deposition have been demonstrated to increase the release of pro-thrombotic cytokines [48]. Finally, the inhibition of CD59 induces the increased production of monocyte chemoattract protein-1 (MCP-1) with consequent monocyte activation and migration in the sub-endothelial layer [49]. 

### 2.5. Platelets

Chronic hyperglycaemia and GV have been reported to promote platelet activation and aggregation [50]. Firstly, hyperglycaemia is associated with a significant reduction in endothelial NO production. The underlying mechanisms have been previously reported; thus, hyperglycaemia-induced superoxide overproduction inhibits eNOS and activates PKC and NF-kB with subsequent ROS production. NO is fundamental to reduce platelet aggregation and stimulate preformed platelet aggregates’ disaggregation [51].

Furthermore, the activation of transcription factors such as NF-kB results in increased production of inflammatory chemokines, thus increasing the exposure of endothelial adhesion molecules that favour platelet adherence and aggregation [52]. Increased oxidative stress also activates platelets through augmented response to agonists mediated by increased F2-isoprostane production and decreased prostacyclin levels [53]. Thus, low levels of prostacyclin have been associated with decreased aspirin effectiveness [54,55]. Finally, high blood glucose levels stimulate the expression of tissue factor (TF) by endothelial cells and monocytes.

Glycaemic levels also significantly influence the activity of several platelet surface glycoprotein (GP) receptors. Therefore, increases in GPIa/IIa (collagen receptor), GPIIb/IIIa (fibrinogen receptor), and GPIb-IX (von Willebrand factor receptor) have been reported in the setting of hyperglycaemia [56]. Furthermore, glucose variations stimulate in vivo and in vitro P-selectin (CD62p) production, a lectin family member that mediates platelets rolling on endothelial cells and stabilizes initial aggregates induced by GP IIb-IIIa and fibrinogen (40,41). Platelets of diabetic patients also show increased arachidonic acid-thromboxane (TxA2) production and CD-40 ligand (CD40L) expression, which enhances the production and release of pro-inflammatory cytokines and favours the development of platelet-rich thrombi [56]. Furthermore, platelet Ca^2+^ ATPase activity and intracellular Ca^2+^ signalling are abnormal in hyperglycaemic conditions, increasing platelet reactivity [57].

## 3. Abnormal Glucose Levels and PCI Complications

Patients with DM experience worse outcomes following coronary stenting. Type 1 and type 2 DM are both associated with an increased risk of cardiovascular diseases. This elevated risk is higher in patients with type 1 DM; this might reflect a longer duration of diabetes and different pathogenetic mechanisms, including autoimmune pathways [58]. However, the incidence of type 1 DM is hugely lower in patients with cardiovascular diseases undergoing PCI and in the general population compared with type 2 DM [59]. This explains why most data on diabetes impact in PCI cohorts regards to type 2 DM.

Moreover, several experimental and clinical studies reported that pre-PCI abnormal glycaemic values correlate with procedural complications and long-term outcomes, irrespective of a previous diagnosis of diabetes and particularly in the setting of ACS (Figure 2) [60,61]. Therefore, the term stress hyperglycaemia has also been introduced, referring to acute glucose elevations as part of the stress response in acute critical conditions, such as myocardial ischemia. Notably, high GV appears even more deleterious than sustained hyperglycaemia in patients receiving percutaneous revascularization [62].

### 3.1. No-Reflow Phenomenon

No-reflow is a rare PCI complication associated with increased mortality [62]. It is defined as inadequate myocardial perfusion without angiographic evidence of mechanical vessel obstruction. The underlying pathophysiological mechanisms are not completely known; however, several studies suggest that an abnormal glycaemic control may promote this complication, particularly in the setting of an increased thrombotic milieu such as in patients with ACS. Indeed, Ikawura et al. found a significant correlation between admission glucose levels and the occurrence of the no-reflow in patients with acute myocardial infarction (AMI) undergoing successful percutaneous reperfusion [63]. Afterwards, a large meta-analysis of 27 retrospective and prospective studies confirmed these findings [64]. Finally, another study by Liu et al. demonstrated that higher in-hospital peak glycaemia was an independent predictor of the no-reflow phenomenon in patients undergoing primary PCI [65]. 

Several mechanisms could explain the increased risk of no-reflow in hyperglycaemic conditions. Firstly, hyperglycaemia induces enhanced expression of P-selectin and other pro-thrombotic adhesion molecules on the coronary capillaries, which favours the trapping of leukocytes [66]. Secondly, high glucose levels stimulate platelet activation, aggregation and consequent microthrombus formation in small coronary vessels that could further produce microvasculature damage and the no-reflow phenomenon [67]. Acute hyperglycaemia may also alter the protection afforded by ischemic preconditioning, probably through the attenuation of mitochondrial ATP-regulated potassium channels, that are crucial to protect the myocardium from a prolonged ischemic insult [68,69]. Finally, hyperglycaemia inhibits vessel collateral formation resulting in increased myocardial damage [68]. 

### 3.2. Preprocedural Myocardial Injury

Several investigations demonstrated the predictive role of pre-PCI glucose levels on periprocedural myocardial injury. Notably, this complication has been associated with increased rates of cardiac adverse events on long-term follow-up [70]. Therefore, patients with DM show greater periprocedural myocardial damage than non-diabetics [71]. Xia et al. reported that high GV correlated with an increased incidence of troponin release after PCI [72]. More recently, patients with an abnormal glucose metabolism detected by oral glucose tolerance testing documented a higher incidence of myocardial infarction and cardiac death within 48 h after PCI [73].

In addition, the risk of periprocedural myocardial damage is also increased by hypoglycaemia before or during PCI. Indeed, hypoglycaemia was demonstrated to be an independent predictor of periprocedural injury (OR = 2.53, 95% CI 1.09–5.81) and long-term major adverse cardiovascular events (MACEs) in a cohort of patients undergoing elective PCI [74]. Concordantly, hypoglycaemic variations assessed by continuous glucose monitoring have been observed to increase in patients with PMI [75].

Platelet hyperreactivity, endothelial dysfunction, increased inflammation and oxidative stress are assumed to be the underlying pathophysiological mechanisms of periprocedural myocardial damage, inducing coronary vasoconstriction and microembolization. Since hyperglycaemia, increased GV, and hypoglycaemia might amplify all of these mechanisms, the link between abnormal glucose levels and procedural myocardial damage seems plausible.

### 3.3. Stent Thrombosis

ST is an insidious complication after stent implantation, with most patients presenting with AMI [76]. 

Several clinical investigations reported the direct association between ST and an abnormal glycaemic status. In a large registry, ST occurred more frequently in diabetic patients undergoing PCI than in those without diabetes [77]. In another large multicenter study, diabetes was also associated with an increased rate of acute ST after DES implantation [78]. Iqbal et al. confirmed DM as an independent predictor of definite and probable ST [79]. Notably, this complication is even more frequent in insulin-treated diabetic patients [80]. This may happen due to the fact that these patients are at higher risk of hypoglycaemia, have a worse glycaemic control and are at a more severe stage of the disease. 

Poor glycaemic control can also lead to the recurrence of ST. A retrospective study examined the clinical outcomes among 243 consecutive patients who suffered from ST after the index PCI. The endpoint was 1-year cardiac death or recurrent ST after the initial ST. According to this study, patients with recurrent events had higher blood glucose and HbA1c levels than those without, indicating that glycaemic control is a key factor in cardiovascular risk [81].

Hyperglycaemia by itself has also been correlated with ST, regardless of diabetes. In a study conducted by Zhang et al., patients without DM who suffered from ST-elevation myocardial infarction (STEMI) with stress hyperglycaemia, defined as blood glucose >180 mg/dl measured on admission, experienced a significantly higher incidence of ST than patients with normal glycaemic values [82].

### 3.4. Intra-Stent Restenosis and Neoatherosclerosis

Type 2 DM is an independent predictor of ISR after coronary angioplasty, even doubling its risk. Moreover, long-term type 1 DM enhances ISR development after endovascular stenting in pre-clinical studies [83].

Pre-procedural abnormal glycaemic values have been associated with increased incidence of ISR and target vessel revascularization in diabetic and not diabetic patients undergoing elective PCI [84]. In another two studies, patients with poor glycaemic control on long-term follow-up, defined as higher HbA1c variability, reported a greater ISR incidence after PCI with DES [85]. Finally, pre-procedural optimal glycaemic control was associated with a lower stent failure rate.

Some key processes might contribute to ISR and neoatherosclerosis development in diabetic patients, such as increased VSMC proliferation, enhanced macrophage infiltration, and greater vasa vasorum neovascularization leading to intraplaque haemorrhage [85]. Moreover, the endothelial dysfunction and the consequent impaired NO production caused by impaired glycaemic control and insulin resistance are independent predictors of early restenosis after PCI, as shown in a study conducted by Piatti et al. [86]. Albeit the introduction of DES and the subsequent decrease in the incidence of ISR compared to bare metal stents [87], no significant interaction between the diagnosis of diabetes and the two stent types was found [88]. Finally, increased production of AGEs has been demonstrated to contribute significantly to restenosis pathogenesis. Thus, elevated AGE levels predict ISR occurrence after DES implantation in diabetic patients on optimal glycaemic control and regardless of Hb1Ac values [89].

## 4. Glycaemic Control during Percutaneous Coronary Revascularization

Given the solid pathophysiological basis, efforts have been concentrated over the years to investigate the benefit of optimal glycaemic control during and after percutaneous coronary revascularization. In this regard, Marfella et al. [90] showed that periprocedural tight glycaemic control halves the incidence of ISR at six-month follow-up in diabetic patients with STEMI undergoing PCI. Interestingly, in this study, intensive glycaemic control was also associated with reduced inflammatory and oxidative stress markers compared with patients in the conventional therapy group [90]. 

While periprocedural hyperglycaemia is undoubtedly related to adverse short- and long-term outcomes after PCI, the best strategy to treat hyperglycaemia during the hospital stay is still uncertain. 

Some clinical trials investigated the effects of glucose-insulin-potassium (GIK) therapy, which was proposed for the first time in the 1960s. The assumption was that GIK infusion might have an antiarrhythmic effect through potassium uptake and a beneficial effect on ischemic myocardium by reducing free fatty acid levels and enhancing myocardial efficiency by providing high-dose glucose [91,92]. However, no significant benefit has been reported with GIK infusion in most cases. The CREATE ECLA and GIPS randomized trials found no significant benefit of GIK infusion in AMI patients undergoing primary PCI [93,94]. Moreover, a possible injury with GIK therapy has been proposed; a meta-analysis of 38 randomized trials enrolling patients undergoing coronary revascularization by PCI or coronary artery bypass surgery demonstrated increased mortality in patients receiving periprocedural GIK infusion [95]. Notwithstanding, the lack of strict glucose monitoring and an increased risk of hypoglycaemic episodes during the study might explain these findings.

Insulin-based strategies for periprocedural glycaemic control have been largely evaluated with conflicting results. The DIGAMI 1 trial showed a mortality rate decreasing about 30% after one year and an absolute reduction in mortality of 11% in the long-term follow-up (3–5 years) in patients with AMI treated with insulin-glucose infusion followed by subcutaneous insulin [96]. The DIGAMI 2 trial confirmed the glycaemic decompensation in the acute phase of myocardial ischemia as an independent predictor of long-term mortality in patients presenting with ACS [97]. Subsequently, the NICE-SUGAR study compared intensive versus conventional glucose control in critically ill patients [98]. This study showed a significant increase in mortality rate at 90 days in the intensive glycaemic control group due to a high incidence of severe hypoglycaemia (glycaemia ≤ 40 mg/dL) compared with patients receiving standard protocols [98]. Similarly, a meta-analysis of 26 trials, including the NICE-SUGAR study, concluded that tight glycaemic control with an insulin-based strategy increases the risk of hypoglycaemia with no overall mortality benefit among critically ill patients [99]. 

Therefore, optimal glycaemic control in the setting of percutaneous coronary revascularization should take into account three different domains: hyperglycaemia, hypoglycaemia, and GV. Insulin is the only treatment studied in depth; advantages of insulin treatment include the rapid corrections of blood glucose and its use in critically ill patients who cannot eat regular meals [100]. However, insulin is often associated with a significant increase in the risk of severe hypoglycaemia, which should be avoided in diabetic patients. Oral glucose-lowering agents could represent an alternative strategy, particularly in patients already treated with them at hospital admission. Nevertheless, some of these drugs should be avoided or stopped in PCI patients. 

Thiazolidinediones showed anti-atherosclerotic effects but exacerbated HF by inducing fluid retention [101,102]. These agents (rosiglitazone, pioglitazone) have demonstrated to decrease the rate of restenosis and the need for target vessel revascularization in patients undergoing PCI [103]. On the other hand, the potential role of sulfonylureas is more controversial due to the high risk of hypoglycaemia associated with these molecules [104]. Kao et al. showed that metformin therapy was associated with a reduced risk of AMI and mortality in diabetic patients undergoing PCI [105]. Although metformin does not raise the risk of hypoglycaemia, it has been demonstrated to enhance the risk of lactic acidosis in patients with kidney disease, a rare but dangerous complication with a mortality of 30–50% [106]. There is also a hypothetical risk of deteriorating renal function in patients treated with metformin exposed to iodinated contrast media and who have pre-existing kidney disease. Indeed, current guidelines on myocardial revascularization recommend checking renal function after coronary angiography for at least 72 h and discontinuing metformin 48 h before angiography/PCI when renal function is deteriorated [107].

In this background, novel anti-diabetic agents, like dipeptidyl peptidase-4 inhibitors (DPP-4i), glucagon-like peptide-1 receptor agonists (GLP-1 RAs) and sodium–glucose cotransporter-2 inhibitors (SGLT-2i) seem to be particularly attractive due to the low risk of hypoglycaemia and the potential beneficial effects on cardiovascular outcomes reported in large, randomized trials on patients with CAD regardless of diabetes [100].

## 5. Novel Anti-Diabetic Drugs in the Setting of PCI

### 5.1. Glucagon-Like Peptide-1 Agonists

#### 5.1.1. Mechanisms of Action

GLP-1 is a gut-derived peptide which increases glucose-stimulated insulin secretion in response to nutrient intake [108]. Since the introduction of exenatide in 2005, several GLP-1 RAs have been approved for use in diabetic patients, with improved effectiveness and longer duration. Notably, gastrointestinal disorders, such as nausea, diarrhoea and weight loss, are the most common adverse reactions associated with these drugs; less frequently, acute pancreatitis might be reported. 

Albeit the mechanisms responsible for the cardiovascular effects of these agents are still unclear, a direct cardiac and vascular action has been reported beyond the improvement of glycaemic control (Figure 3). GLP-1 RAs may reduce chronic cardiac inflammation, which affects diabetic hearts resulting in progressive evolution to dilated cardiomyopathy [90]. They also exert their beneficial effects through raised autophagy, decreased inflammatory protein release and ROS production [91]. Furthermore, GLP-1 RAs oppose the hyperglycaemia-induced apoptosis in cardiomyocytes by suppressing RAGE activity and expression [92].

GLP-1 causes vasodilation and improves endothelial function [108]. Indeed, Oeseburg et al. [108] demonstrated that GLP-1 reduces the ROS-induced senescence of in vitro endothelial cells through mechanisms involving downstream protein kinase A signalling and induction of antioxidant genes. Similarly, the GLP-1 RA exendin-4 provides resistance to angiotensin II-induced O2– generation, thus reducing the ageing and senescence of VSMCs [109,110]. In addition, liraglutide showed a protective effect on endothelial function and oxidative stress through the rapamycin (mTOR) pathway, which promotes NO availability and reduces cell apoptosis by downregulating proapoptotic pathways and upregulating anti-apoptotic proteins [111]. GLP-1 and its analogues also reduce intracellular ROS, preventing oxidative stress damage and promoting endothelial cells and VSMCs protection [112]. 

In addition, several studies also demonstrated the antiatherogenic activity of GLP-1 RAs independently of their effect on glycaemic stabilization [113]. Specifically, GLP-1 prevents the progression of atherosclerotic lesions by inhibiting the formation of macrophage foam cell clusters [114]. Concordantly, liraglutide has been reported to prevent the development of atherosclerotic plaques through the suppression of macrophage foam cell formation by down-regulation of acyl-CoA:cholesterol acyltransferase 1 (ACAT1) [115], responsible for cholesterol ester accumulation [116]. Semaglutide, another GLP-1 RA, showed to decrease inflammation and pro-atherogenic mechanisms through the reduction of cytokine levels (i.e., TNF-α and interferon γ [IFN-γ]) and immune cells activation, recruitment, and adhesion [117]. 

Furthermore, GLP-1 RAs provide a direct cardioprotective effect, increasing the β-catenin signalling and canceling apoptosis [118]. GLP-1 RAs demonstrated a suppressive effect on mitochondrial dysfunction and oxidative stress by inhibiting methylglyoxal, an intermediate of advanced glycation end-products [119]. 

#### 5.1.2. Pre-Clinical and Clinical Evidence 

Several randomized trials have examined the effects of GLP-1 RAs on cardiovascular outcomes in patients with type 2 diabetes and high cardiovascular risk profile, showing a significant benefit, particularly for liraglutide (LEADER) [120], albiglutide (Harmony outcomes) [121], semaglutide (SUSTAIN-6) [122] and dulaglutide (REWIND) [123]. A recent meta-analysis, including seven large-scale trials (ELIXA, LEADER, SUSTAIN-6, EXSCEL, Harmony outcomes, REWIND and PIONEER-6), has confirmed the benefit of GLP1-RAs on cardiovascular outcomes: treatment with GLP-1 RAs reduced MACE (a composite endpoint consisting of cardiovascular death, non-fatal stroke, non-fatal MI) by 12% (hazard ratio [HR] 0.88, 95% confidence interval [CI] 0.82–0.94; *p* < 0.0001) and it was associated with a significant reduction of all-cause mortality, hospital admission for HF, and a composite kidney outcome (development of new-onset macroalbuminuria, decline in estimated glomerular filtration, progression to end-stage kidney disease, death attributable to kidney causes) [124].

However, the effects of GLP1-RAs in the specific setting of PCI are still not completely clarified. Some preclinical studies have shown a cardioprotective action of GLP-1 RAs in animal models of myocardial ischemia. For instance, Timmers et al. investigated the effects of exenatide treatment after 75 min of coronary artery ligation and subsequent reperfusion in a pig model [125]. In this study, exenatide reduced infarct size by 40% and significantly improved systolic and diastolic cardiac function. Similarly, in another study, liraglutide decreased infarct size in a murine ischemia model by activating prosurvival kinases and cytoprotective pathways [126].

Clinical studies also confirmed the potential benefit of GLP1-RAs in reducing infarct size in STEMI patients undergoing PCI [127,128]. Therefore, Lønborg et al. showed that administration of exenatide at the time of reperfusion was associated with an improvement in myocardial salvage and a reduction of infarct size [127]. Concordantly, Woo et al. enrolled 58 STEMI patients undergoing primary PCI, randomized to receive either exenatide or placebo [128]. Infarct size was evaluated by measuring creatine kinase-MB and troponin I blood levels and performing cardiac magnetic resonance one month after AMI. Exenatide significantly reduced creatine kinase-MB and troponin I levels than placebo. Cardiac magnetic resonance showed a smaller area of delayed hyperenhancement in the exenatide group compared to the control group. An echocardiographic exam performed six months after AMI revealed a significantly lower value of E/E’ and improved strain parameters in the exenatide group. 

Similar findings have been observed for liraglutide, which improves myocardial salvage and reduces infarct size after STEMI, probably by attenuating reperfusion injury [129]. Thus, Chen et al. found that treatment with liraglutide was associated with improved left ventricular ejection fraction three months post revascularization in STEMI patients undergoing primary PCI [130]. In addition, the same authors found that liraglutide significantly lowered the incidence of no-reflow in STEMI patients compared with placebo [131]. 

Finally, several experimental investigations demonstrated that GLP1-RAs suppress ISR by inhibiting VSMCs migration and proliferation [132,133]. Exendin-4 attenuated neointimal hyperplasia (NIH) progression after vascular injury in a mice femoral-artery model by reducing platelet-derived growth factor-B (PDGF-B) induced VSMCs proliferation [134]. Liraglutide dose-dependently blocked NIH in vivo [135]. Interestingly, this effect was nullified by the NOS inhibitor N-omega-nitro-L-arginine methyl ester (L-NAME), proposing that liraglutide exerts its anti-restenotic effects also through an increased endothelial NO production [135]. In diabetic pigs treated with liraglutide, this agent reduced NIH after stent implantation via regulation of GV, the NLR family pyrin domain containing 3 (NLRP3) inflammasome, and interleukin 10 (IL-10) levels [136]. However, despite these exciting findings, further clinical studies are needed to investigate whether GLP-1 RAs might represent a potentially effective pharmacological therapy to prevent ISR in patients undergoing PCI.

### 5.2. Dipeptidyl Peptidase-4 Inhibitors

#### 5.2.1. Mechanisms of Action

DPP-4i (alogliptin, linagliptin, vildagliptin, saxagliptin, sitagliptin) suppress the breakdown of incretin hormones GLP-1 and glucose-dependent insulinotropic peptide, improving glycaemic control. DPP-4i also showed beneficial effects on inflammatory markers, oxidative stress, and endothelial function in patients with DM, exhibiting enzymatic activity against dozens of peptide hormones and chemokines involved in different molecular pathways (Figure 3) [137]. These agents have been well-tolerated in both short- and long-term studies; the most common adverse reactions were upper respiratory tract infection, nasopharyngitis, and headache. These agents have also been associated with an increased risk of inflammatory bowel disease [138]. 

Concerning their cardiovascular effects, previous studies showed that DPP-4i reduce the expression of several adhesion molecules on endothelial cells (i.e., E-selectin, ICAM-1, and VCAM-1), preventing atherosclerotic plaque progression and potentially stent-related complications such as ST and ISR [139,140]. In this regard, vildagliptin has been reported to improve endothelial function in mice stimulating NO production and release [141]. Linagliptin decreases ROS production promoted by AGE and NOX expression, thus reducing vascular oxidative stress [140].

Furthermore, several studies suggest anti-inflammatory properties for DPP-4i [142]. Sitagliptin reduces the expression of pro-inflammatory markers (IL-6 and TNF-α) and circulating molecule levels (E-selectin and C-reactive protein) [143,144]. This DPP-4i agent also promotes atherosclerotic plaque stability by increasing collagen synthesis and reducing the expression and activity of metalloproteinases [145]. Furthermore, the anti-inflammatory effects of DDP-4i rely on the reduction of activation and migration of macrophages and T lymphocytes in atherosclerotic plaques and on the raised levels of the anti-inflammatory molecule IL-10 [146]. DPP-4i have also been demonstrated to increase the mobilization and activity of circulating endothelial progenitor cells (EPCs), which play a crucial role in angiogenesis and endothelial repair [22]. Finally, several clinical and pre-clinical studies indicated the anti-thrombotic effects of DPP-4i mediated by mechanisms such as attenuation of mitochondrial respiration and thrombin-induced aggregation, increased intracellular cyclic AMP levels improving NO availability, inhibitory effects on tyrosine phosphorylation and intracellular free calcium [137,147,148].

#### 5.2.2. Pre-Clinical and Clinical Evidence

Several randomized clinical trials have assessed the cardiovascular effects of DPP-4i in diabetic patients. Although the experimental evidence of anti-inflammatory and anti-atherogenic effects of DPP-4i, none of the trials showed a significant benefit on cardiovascular outcomes [149,150,151,152,153]. Moreover, the SAVOR-TIMI 53 trial highlighted an unexpected increased risk of hospitalizations for HF with saxagliptin [149]. However, most of these trials were designed to investigate the non-inferiority of this class of new antidiabetic agents over placebo rather than the superiority, thus testing their cardiovascular safety. 

Few studies focused on the potential role of DPP-4i in the setting of coronary revascularization. Some experimental analyses evaluated the effects of DPP-4i on the processes of post-angioplasty reendothelization, considering that the regenerated endothelium is usually dysfunctional in diabetic patients and strictly correlated to ISR, neo-atherosclerosis and late ST development [154]. In a mice model, a 16-week treatment with anagliptin significantly attenuated the accumulation of monocytes and macrophages in the vascular wall; moreover, it reduced migration and proliferation of VSMCs in plaque areas by inhibiting ERK phosphorylation [155]. An interesting study by Lee et al. investigated the effects of a vildagliptin eluting stent in vitro and in vivo in diabetic rabbits [156]. The results showed that vildagliptin eluting stents promoted arterial healing after stent deployment and reduced VSMCs hyperplasia [156]. 

In humans, there is limited evidence about the effects of DPP-4i on PCI related outcomes. A retrospective analysis, including data from the Acute Coronary Syndrome Israeli Survey (ACSIS) 2010, showed that chronic treatment with sitagliptin was associated with a lower risk of in-hospital complications and 30-day MACEs among diabetic patients presenting with ACS [157]. In-hospital complications included acute renal failure, pulmonary oedema and infections, while 30-day MACEs included either ST, urgent revascularization, post-event ischemia, 30-day mortality, re-infarction or re-ischemia, re-admission and stroke/transient ischemic attack. Another observational study investigated the association between incretin treatment (GLP-1 RAs and DPP-4i) and stent-related complications. In contrast with the ACSIS study, among 18505 diabetic patients undergoing PCI with DES, GLP-1 RAs and DPP-4i did not reduce the risk of ST and ISR compared with patients not receiving these drugs [158]. 

### 5.3. Sodium–Glucose Cotransporter-2 Inhibitors

#### 5.3.1. Mechanisms of Action

SGLT-2i inactivate sodium-glucose cotransporters located in proximal renal tubules, thus inducing sodium and glucose excretion to improve glycaemic control [159]. However, since the cardiovascular benefit of SGLT-2i is mainly not associated with their glucose-lowering effect and arises too early to seem a consequence of weight reduction, direct vascular effects have been invoked (Figure 3). Therefore, empagliflozin has been demonstrated to attenuate oxidative stress and improve endothelial function in diabetic rats by inhibiting AGEs and inflammation [160]. SGLT-2i therapy decreases the up-regulation of the mRNA encoding for eNOS and P-selectin [160]. In addition, SGLT-2 inhibition reduces ROS production by lowering mitochondrial oxidative metabolism, reducing glucotoxicity, and favouring anti-inflammatory processes [161]. 

Dapagliflozin also decreases cardiac fibrosis by activating anti-inflammatory cells and attenuating myofibroblast activation after myocardial injury in mice [162]. Similarly, empagliflozin inhibits myofibroblast activation and differentiation, preventing cell-mediated extracellular matrix collagen remodelling [163]. Furthermore, a recent study demonstrated lower platelet activation markers in patients treated with dapagliflozin and empagliflozin [164]. SGLT-2i could attenuate platelet aggregation by reducing ROS production and release and enhancing NO bioavailability [139]. Finally, treatment with SGLT-2i could promote vasorelaxation and improve endothelial function [165,166]. Gaspari et al. [164] showed that acute dapagliflozin treatment could induce endothelium-independent vascular relaxation; chronic administration of the same drug improved endothelial function and decreased the expression of adhesion molecules, thus preventing macrophages infiltration [164].

Despite these beneficial pleiotropic properties, some critical side effects should be mentioned for this class of agents. Further to the most common genitourinary tract infections, volume depletion and haemoconcentration induced by these drugs raised the question of a possible ischemic and pro-thrombotic effect. Hence, these mechanisms have been called into question to explain the increased risk of peripheral ischaemia with limb amputation and acute kidney injury observed in patients receiving SGLT-2i [167,168]; concerns that were disproved by several investigations. Indeed, some authors observed that the hematocrit rise triggered by these agents might promote oxygen delivery to the myocardium and reduce the workload of the proximal renal tubules, improving tubule-interstitial hypoxia [169]. Of note, another feared side effect of SGLT-2i is represented by the euglycemic diabetic ketoacidosis, for which some regulator warnings have been emanated.

#### 5.3.2. Pre-Clinical and Clinical Evidence 

Four large clinical trials have investigated the effects of SGLT-2i in patients with type 2 diabetes and high cardiovascular risk: EMPA-REG OUTCOME for empagliflozin, CANVAS for canagliflozin, DECLARE-TIMI 58 for dapagliflozin, VERTIS-CV for ertugliflozin [166,167,170,171]. Those trials generated great enthusiasm since they showed, on the one hand, the superiority of empagliflozin and canagliflozin over placebo in reducing the incidence of MACEs and, on the other hand, the reduction of HF hospitalizations for all of the five agents. The beneficial effects of SGLT-2i in patients with HF are furtherly supported by recent superiority trials, such as DAPA-HF and EMPEROR-reduced [8,172]. Moreover, SGLT-2i improve renal outcomes, regardless of pre-existing cardiovascular disease [173,174,175].

Furthermore, the anti-inflammatory, antioxidant and antithrombotic properties of this class of agents, along with favourable alterations in lipid metabolism, may support the role of SGLT-2i in preventing stent-related complications. However, there is limited evidence about how SGLT-2i influence outcomes in patients receiving PCI.

Some pre-clinical and clinical studies have suggested that SGLT-2i could affect vascular remodelling and attenuate NIH after angioplasty. For instance, an experimental study conducted by Mori et al. showed that luseogliflozin attenuated NIH of the femoral artery after wire injury in high-fat diet-fed mice by suppressing the macrophage PDGF-B [176]. Similar findings have been observed in humans. A single-centre prospective study investigated the effect of empagliflozin on NIH after DES implantation in patients with type 2 diabetes [177]. The primary endpoint was the thickness of in-stent NIH at 12 months after PCI, assessed by optical coherence tomography (OCT). Interestingly, the OCT analysis showed that the thickness of NIH was significantly lower in the empagliflozin group than in the control group [177]. 

As GLP-1 RAs, even SGLT-2i were found to reduce infarct size in experimental ischemia/reperfusion injury models. Specifically, long-term treatment with empagliflozin reduced myocardial infarction size and improved myocardial function in mice with type 2 diabetes subjected to 30 min of ischemia and 2 h of reperfusion [178]. An additional preclinical study demonstrated that dapagliflozin lowers cardiac infarct size, increases left ventricular function and decreases arrhythmias in rats with cardiac ischemia/reperfusion injury [179]. 

Among the new glucose-lowering agents, SGLT-2i have been proved to have potential nephroprotective effects. Since contrast-induced acute kidney injury (CI-AKI) causes clinically acquired nephropathy and worsens pre-existing nephropathy in PCI patients, it is reasonable to ask if SGLT-2i could prevent periprocedural renal complications. In this regard, an experimental study on rats demonstrated that dapagliflozin attenuates CI-AKI by suppression of hypoxia-inducible factor 1-alpha (HIF-1α)/human epididymis protein-4 (HE4)/NF-κB signalling, which is related to hypoxic injury [180].

## 6. Relevant Drug Interactions of Novel Antidiabetic Agents with Standard Therapy

Patients undergoing PCI require pharmacological therapy including antiplatelet drugs, beta-blockers, statins, and ACE inhibitors. Therefore, possible interactions with this standard therapy should be considered when antidiabetic drugs are administered.

The presence of diabetes mellitus, such as hyperglycaemia and high glycaemic variability, has been associated with high platelet reactivity in patients undergoing coronary stenting, with current antiplatelet agents proven to be less effective in these patients [181]. Of note, several investigations have associated an impaired response to antiplatelet therapy with worse long-term clinical outcomes [182]. 

In this context, some glucose-lowering agents have been reported to reinforce the antithrombotic effects of antiplatelet agents, and others have been shown to counteract the effects of these drugs. Indeed, metformin furtherly decreases platelet activation among diabetic patients on aspirin therapy [183]. Conversely, sulphonylureas are associated with reduced clopidogrel efficacy in inhibiting platelet aggregation measured using light transmittance aggregometry [184]. Moreover, the concomitant use of aspirin with sulphonylureas seems to potentiate the antiplatelet effect of the former and the hypoglycaemic effect of the latter [185,186]. Finally, clopidogrel enhances the effect of pioglitazone, inhibiting its CYP2C8-mediated biotransformation [187]. 

Notwithstanding the well-documented antiplatelet effects of the novel antidiabetic agents, no studies specifically designed to find interactions between GLP-1 RAs, SGLT-2i and DPP-4i with antiplatelet drugs are available to date. Whether on one side, this could indeed represent an interesting topic for further studies; on the other, these drugs have been reported to have critical interactions with other cardiovascular drugs. The DPP-4i vildagliptin was reported to increase the bioavailability of ACE inhibitors, thus potentially increasing the risk of angioedema [188]. GLP-1RAs showed possible interactions with statins and ACE inhibitors [189]. Exenatide slightly reduces lovastatin bioavailability, whereas liraglutide and semaglutide cause a delay in atorvastatin’s time to reach maximum concentration [190]. Recent investigations suggested that concomitant use of SGLT-2i with statins could lead to increased statin exposure and toxicity [191]. On the other hand, diuretics such as torasemide and hydrochlorothiazide might cause an increase in SGLT-2i concentrations [192].

## 7. Limitations and Future Perspectives

In this review, we focused on patients undergoing percutaneous coronary revascularization summarizing the existing evidence on the effects of new antidiabetic drugs in the setting of PCI. We have not considered the effects of these drugs on patients undergoing surgical revascularization through coronary artery bypass graft (CABG). However, it is well known that CABG is a relevant treatment option in diabetic patients with CAD, being superior to PCI with DES in reducing the rate of MACE in patients with DM and multivessel disease in several randomized trials [193]. Indeed, previous investigations have already demonstrated the potential benefit of SGLT-2i and GLP-1 RAs in those patients [194,195].

Additionally, we included all of the evidence in the PCI setting without differentiating the acute from the elective revascularization. Patients with ACS are more likely to experience stress hyperglycaemia, regardless of a previous history of known diabetes. Notably, stress hyperglycaemia is associated with worse short- and long-term prognosis post PCI [196]. Therefore, it could be interesting to evaluate the different impact of new antidiabetics in patients with ACS and chronic coronary syndrome treated with PCI, probably expecting an even more detectable benefit in the acute setting for the previously mentioned pathophysiological mechanisms. Moreover, most of the data mentioned in our review come from experimental studies conducted on animal models (Table 1). Few data are available from clinical studies, indicating the need for future clinical works.

Finally, notwithstanding all our efforts to summarize all available data on this topic carefully and thoroughly, as all review articles, also our manuscript might be subject to potential bias, such as gaps in literature searching, and mistranslation or misinterpretation of data from the primary literature.

## 8. Conclusions

The strict association between abnormal glycaemic values and the progression of cardiovascular disease is well-known and widely investigated so far. An abnormal glycaemic status, regardless of the presence of DM, has also been related to poor outcomes during and after PCI. Pathophysiologic mechanisms by which glucose abnormalities may be a causal factor for procedural and long-term complications in patients undergoing coronary stenting include endothelial dysfunction, oxidative stress, inflammation, and platelet activation. Given the negative role of glucose abnormalities in patients receiving stent implantation, considerable attention has been focused on determining whether optimal glycaemic control may lead to improved cardiovascular outcomes in these patients. Novel antidiabetic agents such as GLP-1 RAs, DPP-4i, and SGLT-2i significantly improve glycaemic parameters and exert direct protective effects on endothelial and vascular smooth muscle cells through anti-oxidative effects, anti-inflammatory and anti-thrombotic properties, potentially attenuating PMI, ST, and ISR. Pre-clinical studies have provided exciting evidence in this direction, whereas few clinical data are available until now. However, given the increasing use of percutaneous techniques for coronary atherosclerosis treatment in the last years, further studies aiming to examine the role of these new agents in this setting are strongly demanded in order to identify new effective strategies for secondary prevention of patients undergoing PCI.

## Figures and Tables

**Figure 1 ijms-23-07261-f001:**
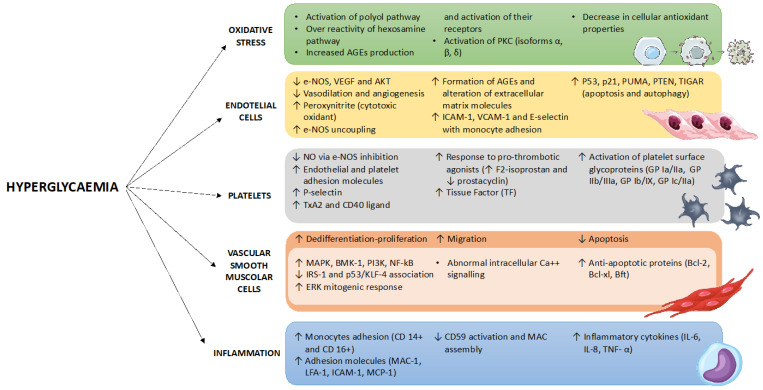
Pathophysiological mechanisms of hyperglycaemia-induced vascular damage. AGE, advanced glycation end products; PKC, protein kinase C; e-NOS, endothelial nitric oxide synthase; VEGF, vascular endothelial grow factor; ICAM-1, intercellular adhesion molecule-1; VCAM-1, vascular cell adhesion molecule-1; PUMA, p53 upregulated modulator of apoptosis; PTEN, phosphatase and TENsin homolog deleted on chromosome 10; TIGAR, TP53-Induced Glycolysis and Apoptosis Regulator; NO, nitric oxide; TxA2, Thromboxane A2; TF, tissue factor; GP, glycoprotein; MAPK, mitogen-activated protein kinase; BMK, big MAPK; PI3K, Phosphoinositide 3-kinases; NF-kB, nuclear factor kappa B; IRS-1, insulin receptor substrate 1; KLF-4, Kruppel Like Factor 4; ERK, extracellular signal-regulated kinase; BCL, B-cell lymphoma; Bft, Bacteroides fragilis toxin; MAC, membrane attack complex; LFA-1, lymphocyte function-associated antigen 1; MCP-1, monocyte chemoattractant protein-1; IL, interleukin; TNF, tumour necrosis factor.

**Figure 2 ijms-23-07261-f002:**
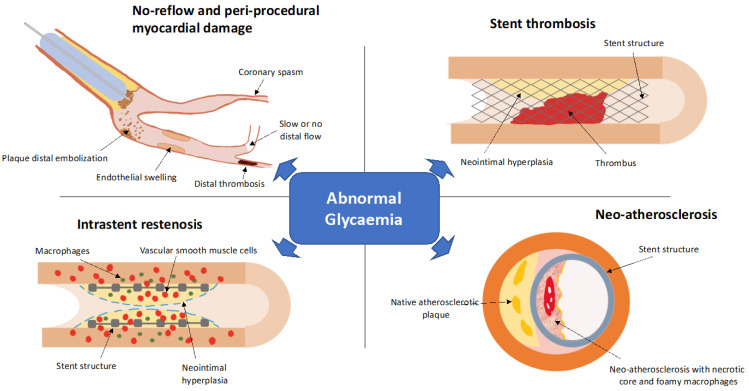
Peri-procedural and long-term stent-related complications favored by an abnormal glycaemic status.

**Figure 3 ijms-23-07261-f003:**
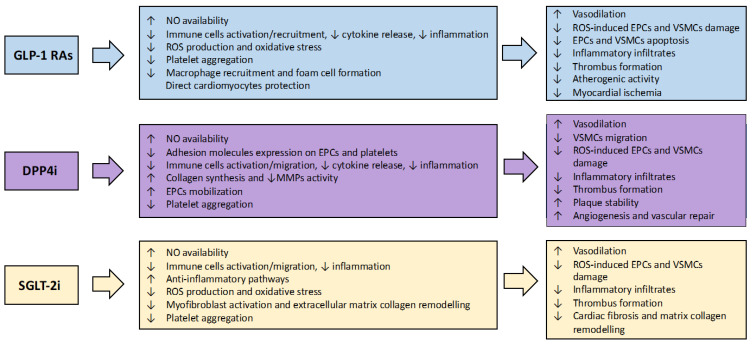
Potential mechanisms of benefit of novel antidiabetic agents in the setting of PCI. PCI, percutaneous coronary intervention; GLP-1 RAs, glucagon-like peptide 1 receptor agonist; DPP4i, dipeptidyl peptidase-4 inhibitors; SGLT-2i, sodium-glucose cotransporter-2 inhibitors; NO, nitric oxide; ROS, reactive oxygen species; EPCs, endothelial progenitor cells; MMPs, matrix metalloprotease; VSMCs, vascular smooth muscular cell.

**Table 1 ijms-23-07261-t001:** Pre-clinical and clinical studies investigating the effects of novel anti-diabetic agents on stent-related complications.

GLP-1 RAs	Type Of-Study	Molecule	Setting	Main Results
Infarct size/periprocedural ischemia				
*Timmers et al., 2009*	Preclinical study	Exenatide	Porcine model of ischemia/reperfusion	↓ infarct size ↑ systolic and diastolic cardiac function
*Ashraf et al., 2009*	Preclinical study	Liraglutide	Induced myocardial infarction in diabetic and non-diabetic mice	↓ infarct size ↓ cardiac rupture ↑ survival ↑ expression and activity of cardioprotective genes (Akt, GSK3 beta, PPAR beta-delta, Nrf-2, and HO-1)
*Lønborg et al., 2012*	Clinical study	Exenatide	Patients with STEMI and TIMI flow 0/1 undergoing primary PCI	↓ infarct size (particularly, in those patients with a short duration of ischemia - ≤132 min)
*Woo et al., 2013*	Clinical study	Exenatide	Patients with STEMI and TIMI flow 0 undergoing primary PCI	↓ infarct size ↑ left ventricular function (lower E/E' and improved strain parameters)
*Chen et al., 2016*	Clinical study	Liraglutide	Patients with STEMI undergoing primary PCI	↑ myocardial salvage index ↓ infarct size ↓ serum CRP
**No reflow**				
*Chen et al., 2015*	Clinical study	Liraglutide	Patients with STEMI undergoing primary PCI	↑ left ventricular function at 3 months post PCI ↓ no reflow ↓ stress hyperglycaemia
*Chen et al., 2016*	Clinical study	Liraglutide	Patients with STEMI undergoing PCI	↓ no reflow ↓ serum CRP at 6 h post PCI
**ISR**				
*Shi et al., 2015*	Preclinical study	Liraglutide	VSMCs from rat thoracic aorta	↓ migration and proliferation of VSMCs Inhibition of PI3K/Akt and ERK1/2 signaling pathways
*Hirata et al., 2013*	Preclinical study	Exendin-4	Vascular injury in C57BL/6 mice	↓ neointima hyperplasia
*Xia et al., 2020*	Preclinical study	Liraglutide	Diabetic pigs undergoing DES implantation	↓ neointima hyperplasia via regulation of glycaemic variability, NLRP3 inflammasome and IL-10
**DPP-4i**	**Type of study**	**Molecule**	**Setting**	**Main results**
**ISR**				
*Terawaki et al., 2014*	Preclinical study	Linagliptin	Vascular injury in C57BL/6 mice	↓ neointima hyperplasia ↓ VSMCs proliferation
*Lee et al., 2019*	Preclinical study	Vildagliptin	Nanofibrous vildagliptin-eluting stents in diabetic rats	↓ neointima formation ↓ VSMCs proliferation
**Stent thrombosis/Reinfarction**				
*Leibovitz et al., 2013*	Clinical study	Sitagliptin	Diabetic patients presenting with ACS	↓in-hospital complications ↓ 30-day MACEs (stent thrombosis, urgent revascularization, post event ischemia, 30-day mortality, re-infarction or re-ischemia, re-admission, stroke/TIA)
**SGLT-2i**	**Type of study**	**Molecule**	**Setting**	**Main results**
**Infarct size/periprocedural ischemia**				
*Andreadou et al., 2017*	Preclinical study	Empaglifozin	Murine model of ischemia/reperfusion	In vivo: ↓ infarct size ↑ myocardial function In vitro: ↑ STAT3 expression and activation with antioxidant and anti-inflammatory action ↓ myocardial IL-6 and iNOS
*Lahnwong et al., 2020*	Preclinical study	Dapaglifozin	Murine model of ischemia/reperfusion	↓ infarct size ↓ cardiac apoptosis and ↑ cardiac mitochondrial function ↑ left ventricular function ↓ arrhythmias
**ISR**				
*Mori et al., 2019*	Preclinical study	Luseoglifozin	Femoral artery wire injury in mice	↓ neointima hyperplasia
*Hashikata et al., 2020*	Clinical study	Empaglifozin	Diabetic patients undergoing PCI	↓ neointima hyperplasia
**Contrast-induced acute kidney injury**				
*Huang et al., 2022*	Preclinical study	Dapaglifozin	In vitro hypoxia model; diabetic rats receiving contrast media and exhibiting induced CI-AKI	In vitro: ↓ oxygen consumption, HIF-1α, HE4, NF-κB expression and apoptotic cells In vivo: ↓ serum creatinine, urea nitrogen, TUNEL-positive tubular cells, HIF-1α, HE4, NF-κB expression, and histopathological scores related to CI-AKI

GLP-1 RAs, glucagon-like peptide-1 receptor agonists; STEMI, ST elevation myocardial infarction; TIMI, thrombolysis in myocardial infarction; PPAR, peroxisome proliferator-activated receptor; PCI, percutaneous coronary intervention; HO-1, heme oxygenase-1; CRP, C-reactive protein; PI3K, phosphatidylinositol 3-kinases; DES, drug eluting stents; NLRP3, NLR family pyrin domain containing 3; IL, interleukin; DPP-4i, dipeptidyl peptidase-4 inhibitors; VSMCs, vascular smooth muscle cells; ACS, acute coronary syndromes; MACEs, major adverse cardiac events; TIA, transient ischaemic attack; SGLT-2i, sodium-glucose co-transporter-2 inhibitors; STAT-3, signal transducer and activator of transcription 3; iNOS, inducible nitric oxide synthase; CI-AKI, contrast-induced acute kidney injury; HIF-1, hypoxia-inducible factor 1-alpha; HE4, human epididymis protein-4; NF-kB, nuclear factor kappa-light-chain-enhancer of activated B cells; TUNEL, terminal deoxynucleotidyl transferase dUTP nick end labeling.

## Data Availability

Not applicable.

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
