# Peer review of "Glycaemic Control in Patients Undergoing Percutaneous Coronary Intervention: What Is the Role for the Novel Antidiabetic Agents? *A Comprehensive Review of Basic Science and Clinical Data"

_ijms, 2022, doi:10.3390/ijms23137261_

Round 1

Reviewer 1 Report

The review article entitled- Glycaemic control in patients undergoing percutaneous coronary intervention: which role for novel antidiabetic agents? is an interesting work.

Some suggestions are attached below;

1 . It would be ideal to change the title to "what is the role for the novel antidiabetic agents ? "

2 . Please include a paragraph explaining the limitations of the review 

3 . Minor edits and spell checks required.

Author Response

We would like to thank the Editor and Reviewers for the careful and thorough reading of our manuscript and for the thoughtful comments and constructive suggestions, which helped to improve the quality of this manuscript.

The review article entitled- Glycaemic control in patients undergoing percutaneous coronary intervention: which role for novel antidiabetic agents? is an interesting work.

We appreciate the positive feedback from the Reviewer.

Some suggestions are attached below:

1 . It would be ideal to change the title to "what is the role for the novel antidiabetic agents?"

We thank the Reviewer for his/her suggestion. We modified the title accordingly in this revised version of the manuscript.

2 . Please include a paragraph explaining the limitations of the review.

We thank the Reviewer for this comment. As suggested, we added a paragraph explaining the limitations of our work and potential suggestions for future investigations in this field (paragraph 7, pages 17-18, lines 738-761).

3 . Minor edits and spell checks required.

We carefully checked the revised version of the manuscript looking for language and grammatical errors. A native English also edited the manuscript. We thank the Reviewer for his/her careful reading of our text.

Reviewer 2 Report

In this study, the authors provide an overview of the biological mechanisms underlying hyperglycaemia-induced vascular damage and the actions of new antidiabetic drugs. Also, they summarized existing evidence on the effects of anti-diabetic drugs in the PCI setting, addressing pre-clinical and clinical studies and drug-drug interactions with antiplatelet agents, thus highlighting new opportunities for optimal long-term management of these patients.

1- What the author means by hyperglycemia regardless of diabetes?

2- Section 5: All described drugs are used for T2D patients. Can the authors say something about the setting of PCI or what is beneficial to cardiovascular outcomes for T1D patients?

3- There is a problem in Table 1.

4- The author mentioned drug-drug interactions in the abstract, I failed to see this in the review.

5- In section 4: the author mentioned some of the side effects of the anti-diabetic drugs such as metformin. In this regard, I wonder if there is anything is know about other side effects of the novel anti-diabetic drugs.

6- Conclusion need to be improved more, trying to elaborate more on the future perspective.  

Author Response

We would like to thank the Editor and Reviewers for the careful and thorough reading of our manuscript and for the thoughtful comments and constructive suggestions, which helped to improve the quality of this manuscript.

In this study, the authors provide an overview of the biological mechanisms underlying hyperglycaemia-induced vascular damage and the actions of new antidiabetic drugs. Also, they summarized existing evidence on the effects of anti-diabetic drugs in the PCI setting, addressing pre-clinical and clinical studies and drug-drug interactions with antiplatelet agents, thus highlighting new opportunities for optimal long-term management of these patients.

We are grateful for the Reviewer's consideration of this manuscript, and we also very much appreciate his/her suggestions, which have been very helpful in improving the manuscript.

What the author means by hyperglycemia regardless of diabetes?

With "hyperglycaemia regardless of diabetes", we refer to the presence of high blood glucose levels, regardless of a history of known diabetes, thus without previous and established diabetes mellitus.

We thank the Reviewer for his/her comment. According to this, we replaced the term "hyperglycaemia regardless of diabetes" with "hyperglycaemia regardless of a history of known diabetes" (page 2, lines 49, 62-63).

A transient elevation of blood glucose levels has been reported in patients who have not been diagnosed with diabetes mellitus in several critical conditions such as acute coronary syndromes, stroke, cerebral haemorrhage, infections, and urgent surgery. Different glycemic cut-offs have been used to define acute hyperglycaemia, the so-called stress hyperglycaemia, also according to the diabetic status. We also discussed this concept in the revised version of the manuscript (page 6, lines 246-249).

Notably, acute hyperglycaemia is associated with worse short- and long-term outcomes, especially in patients without known diabetes (Stress-Induced Hyperglycaemia in Non-Diabetic Patients with Acute Coronary Syndrome: From Molecular Mechanisms to New Therapeutic Perspectives, Int J Mol Sci 2021; Stress Induced Hyperglycemia in the Context of Acute Coronary Syndrome: Definitions, Interventions, and Underlying Mechanisms, Front. Cardiovasc. Med 2021). However, in this scenario, whether hyperglycaemia may reflect, to some extent, the underlying critical conditions, it should not be considered a simple marker given the growing evidence of negative effects of hyperglycaemia itself on inflammation, apoptosis, and oxidative stress.

2- Section 5: All described drugs are used for T2D patients. Can the authors say something about the setting of PCI or what is beneficial to cardiovascular outcomes for T1D patients?

Thanks to the Reviewer for his/her helpful comment.

Patients with type 1 diabetes mellitus are undoubtedly characterized by an elevated cardiovascular risk, even higher compared with type 2 DM. The pathophysiological mechanisms involved in premature atherosclerosis observed in those patients are partially similar to those observed in T2DM, such as increased inflammation, oxidative stress, and endothelial dysfunction (Glycemic Control, Cardiac Autoimmunity, and Long-Term Risk of Cardiovascular Disease in Type 1 Diabetes Mellitus. Circulation. 2019). However, other mechanisms, such as recurrent hypoglycaemia, autoimmune pathways and functional abnormalities of lipoproteins might be involved (Cardiovascular disease in type 1 diabetes: A review of epidemiological data and underlying mechanisms. Diabetes Metab. 2020). These findings underline that optimal glycaemic control is essential in reducing cardiovascular risk, particularly in T1DM patients.

However, the incidence of type 1 DM is hugely lower in patients with cardiovascular diseases undergoing PCI and in the general population than in type 2 DM (Differential Impact of Type 1 and Type 2 Diabetes Mellitus on Outcomes Among 1.4 Million US Patients Undergoing Percutaneous Coronary Intervention. Cardiovasc Revasc Med. 2022). Therefore, most data in our review come from PCI cohorts with type 2 DM.

In the revised version of our manuscript, as the Reviewer suggested, we discussed the increased CV risk of type 1 DM and added the available data on IRS in a type 1 DM animal model (page 5-6, lines 236-242; page 8, lines 328-329). Notably, to the best of our knowledge, no studies have assessed the effects of novel antidiabetic agents on stent-related complications in T1DM patients.

3- There is a problem in Table 1.

According to the Reviewer’s comment, we carefully checked Table 1, fixing and correcting type errors.

4- The author mentioned drug-drug interactions in the abstract, I failed to see this in the review.

We thank the Reviewer for this very constructive suggestion. We added a further paragraph in this revised manuscript version aiming to briefly investigate potential interactions between novel antidiabetic agents and other cardiovascular drugs, representing the standard of care for patients undergoing percutaneous coronary revascularization (paragraph 6, page 17, lines 705-736).

5- In section 4: the author mentioned some of the side effects of the anti-diabetic drugs such as metformin. In this regard, I wonder if there is anything is know about other side effects of the novel anti-diabetic drugs.

We thank the Reviewer for this suggestion.

We added the most common and dangerous side effects for each glucose-lowering drug class in this revised manuscript version (page 9, lines 425-427; page 12, lines 532-535; page 13, lines 618-629).

Including these potential limitations might provide the reader with a more comprehensive overview of the benefits and risks of using these agents.

6- Conclusion need to be improved more, trying to elaborate more on the future perspective.  

Thanks for the comment.

Also, according to Reviewer 1’s suggestion, we added a new paragraph including limitations and future perspectives on this topic (paragraph 7, pages 17-18, lines 738-761).

Round 2

Reviewer 2 Report

I do not have any comments.